# Fighting Copycat Agents in
# Behavioral Cloning from Observation Histories

**Chuan Wen**[*1], **Jierui Lin**[*2], **Trevor Darrell**[2], **Dinesh Jayaraman**[3], **Yang Gao**[†124]

[1]Institute for Interdisciplinary Information Sciences, Tsinghua University
[2]UC Berkeley, [3]University of Pennsylvania, [4]Shanghai Qi Zhi Institute

## Abstract

Imitation learning trains policies to map from input observations to the actions that an expert would choose. In this setting, distribution shift frequently exacerbates the effect of misattributing expert actions to nuisance correlates among the observed variables. We observe that a common instance of this causal confusion occurs in partially observed settings when expert actions are strongly correlated over time: the imitator learns to cheat by predicting the expert's *previous* action, rather than the next action. To combat this "copycat problem", we propose an adversarial approach to learn a feature representation that removes excess information about the previous expert action nuisance correlate, while retaining the information necessary to predict the next action. In our experiments, our approach improves performance significantly across a variety of partially observed imitation learning tasks.

## 1   Introduction

Imitation learning is a simple, yet powerful paradigm for learning complex behaviors from expert demonstrations, with many successful applications ranging from autonomous driving to natural language generation [35, 40, 26, 28, 8, 16, 52]. The key idea underlying these successes is straightforward: given a training dataset of demonstrations by an expert, an agent's action policy $\pi$ can be trained by mapping demonstration states $s_t$ to expert actions $a_t$.

Partially observed settings pose a problem for this approach: rather than the full state $s_t$, only observations $o_t$ are available to the agent. As an example, for an autonomous car driving agent with a forward-facing camera, a single frame observation $o_t$ from the camera omits much of the relevant state information for driving, such as the velocities of vehicles in front of the car, and the presence or absence of vehicles by its side. An policy trained to map $o_t$ to actions $a_t$ would thus be severely limited in such settings [13, 5, 19], especially in imitation learning. However, this limitation may in principle be alleviated by a simple fix: rather than training a policy $\pi(a_t|o_t)$, one could train a policy $\pi(a_t|\tilde{o}_t = [o_t, o_{t-1}, o_{t-2}, \cdots])$, accessing past observations to fill in missing state information from the current observation.

In practice, several prior works have reported that imitation from observation histories sometimes performs *worse* than imitation from a single frame alone [51, 26, 6]. To illustrate why this happens, consider the sequence of actions in an expert demonstration when it starts to drive in response to a red traffic light turning green (Figure 1). Assuming an action space with only two actions,

---

[*]Equal contribution.

[†]Work done while at UC Berkeley.

cwen20@mails.tsinghua.edu.cn, jerrylin0928@berkeley.edu,
trevor@eecs.berkeley.edu, dineshj@seas.upenn.edu, gaoyangiiis@tsinghua.edu.cn

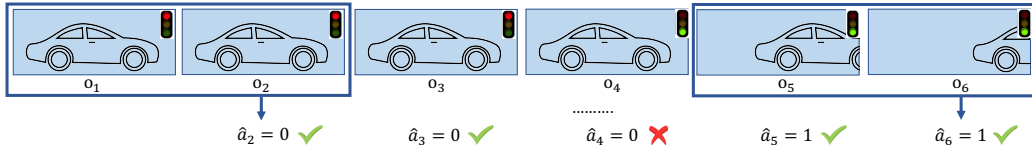

$\hat{a}_2 = 0$ ✓     $\hat{a}_3 = 0$ ✓     $\hat{a}_4 = 0$ ✗     $\hat{a}_5 = 1$ ✓     $\hat{a}_6 = 1$ ✓

Figure 1: This figure demonstrates the "**copycat**" problem in an autonomous driving scenario. The top part of the figure shows a sequence of observations, where the vehicle waits at the red light and start to drive when the light turns green. The policy takes a sliding window of observations as input. At the bottom of the figure, we show that a "copycat" policy which simply replays its previous action will predict all but one actions correctly.

brake ($a = 0$), and throttle ($a = 1$), the sequence of expert actions over time would look like $[a_0 = 0, a_1 = 0, \ldots, a_\tau = 0, a_{\tau+1} = 1, \ldots a_T = 1]$.

Which imitation policies would be effective at predicting the expert action on data from such demonstrations? Consider a "copycat" action policy that simply copies the previous expert action and prescribes repeating it as the next action. On these demonstrations, this policy would produce the correct action at all but one time instant, when the expert switches from braking to throttling. Our imitation learner $\pi(a_t|\tilde{o}_t)$ could easily expresss this copycat policy: it could recover the previous expert action $a_{t-1}$ from the last two frames, $o_t$ and $o_{t-1}$. The imitation training objective would encourage this, since this policy would produce low error on training and held-out demonstrations. However, when testing for actual driving performance, it would be useless — it would simply never switch to the throttle action.

We hypothesize that this copycat problem arises in imitation policies accessing past observations when two conditions are met: (i) expert actions over time are strongly correlated, and (ii) past expert actions are easily recovered from the observation history. As our first contribution, we empirically validate this hypothesis. We find that the temporal correlation among expert actions leads to even higher temporal correlation among learned policy actions. Further, the higher this temporal correlation, the worse the performance of the imitation learner (Section 4).

Then, we propose a novel imitation learning objective with an adversarial learning method that ensures that the imitation policy ignores the known nuisance correlate — the previous action $a_{t-1}$. Our implicit approach is scalable and robust, avoiding the need to learn disentangled representations [7], or learn a mixture of exponentially many graph conditioned policies [12] in previously proposed approaches tackling related problems. Our method only needs an offline expert demonstration dataset, unlike methods like DAGGER [39] and CCIL [12] which resolve the causal confusion through online expert queries, or GAIL [21] which needs online environment interaction. Inspired by robotics applications, we demonstrate our approach in six simulated continuous robotic control settings.

## 2 Related Work

**Imitation Learning.** Imitation learning [30, 2], first proposed by Widrow and Smith in 1964 [53], is a powerful learning paradigm that enables the learning of complex behaviors from demonstrations. We focus on the widely used behavioral cloning paradigm [35, 40, 26, 28, 8, 16], which suffers from a well-known problem: small errors in the learned policy compound over time, leading quickly to states outside the training demonstration distribution, where performance deteriorates. One solution is to assume access to a queryable expert who prescribes actions in the new states encountered by the policy, as in the widely-used DAGGER algorithm [39] and others [46, 25, 47]. Another well-studied alternative is to refine the policy through environment interaction [21, 9].

de Haan et al. [12] explicitly connected distributional shift problems in imitation settings to nuisance correlations between input variables and expert actions, identifying the "causal confusion" problem. We isolate this causal confusion problem in its most frequently occurring form, the copycat problem motivated in Sec 1, encountered by ML practitioners within imitation learning [26, 6, 11, 51] and elsewhere, as "feedback loops" [42, 4]. We demonstrate a scalable solution to the copycat problem.

**Causal Discovery.** While models produced by standard machine learning approaches may rely on nuisance correlates because they assume independent identically distributed samples at training and test time, *causal models* [33, 22, 34] instead uncover relationships that hold even under distributional shift. This connection between causality and distributional robustness has been studied in [20, 27, 10].

Existing causal discovery approaches, such as the widely used PC algorithm [44] as well as more recent techniques [7, 32, 18], operate over predefined, disentangled variables. However, in domains like vision or language, these underlying variables are unknown and must themselves be inferred from raw, high-dimensional observations, making causal discovery hard.

Most closely related to us, in the imitation learning setting, de Haan et al. [12] train a mixture of models on top of learned disentangled features in simple problem settings, exploiting environmental interactions afterwards to find the correct causal model. We take a different approach, sidestepping the difficulties of causal graph learning by injecting domain knowledge [29] to avoid nuisance correlates. Our approach is able to learn policies purely from demonstration data, requiring no environmental interaction afterwards. We compare against de Haan et al. [12] in more detail in later sections.

**Adversarial Learning.** "Adversarial" learning approaches, set up to resemble two agents competing against each other, have recently had great success in many application doamins, such as image generation [17, 36, 23], and domain adaptation [50, 49, 14]. Our approach extends these adversarial losses to the imitation learning setting, setting up a nuisance correlate predictor as an adversary to the imitation policy.

# 3 Background and Problem Setup

**Notation.** We study behavioral cloning (BC) in a partially observed Markov decision process (POMDP). The POMDP states $s_t$ transition as a function of agent actions $a_t$ at time $t$, producing rewards $r_t$ that specify the agent's task within the environment. The agent does not have access to either the states $s_t$, or the rewards $r_t$. In lieu of states $s_t$, it has access to observations $o_t = f(s_t)$, where $f$ is an unknown many-to-one function. In lieu of rewards $r_t$, the agent is provided with an expert's demonstrations of desirable behavior through $N_d$ expert demonstration trajectories $\{\mathcal{T}_i^e\}_{i=1}^{i=N_d}$. BC trains a parametrized policy $\pi_\theta(a_t|\tilde{o}_t)$ that prescribes the next action, given the last $H$ observations $\tilde{o}_t = [o_t, o_{t-1}, o_{t-2}, ...o_{t-H+1}]$. In what follows, we will sometimes refer to observation histories $\tilde{o}$ simply as observations, for brevity.

**BC Training Objective.** The final goal is to maximize the expected cumulative reward $R(\theta) = \mathbb{E}[\sum_t r_t]$ from executing $\pi_\theta$ in the environment. At training time, BC maximizes a different objective: the log-likelihood $L(\theta)$ of a dataset $\{(\tilde{o}_i, a_i^e)\}_{i=1}^N$ of $N$ observation-action pairs drawn from the demonstrations, $\theta^* = \arg\max_\theta \sum_{i=1}^N \log \pi_\theta(a_i^e|\tilde{o}_i)$.

**Distributional Shift.** When the trained policy $\pi_{\theta^*}$ is executed in the environment, even minor deviations from the expert's policy get compounded over time [39]. As a result, $\pi_{\theta^*}$ soon encounters observations $\tilde{o}_t$ outside the training distribution, where its performance suffers. This distributional shift issue lies at the very core of the "copycat" problem studied in this paper.

# 4 The Copycat Problem

To introduce the copycat problem, we start by asking: what is the optimal value of $H$, the size of the observation history window, in the above setup? Conventional wisdom holds [30] that larger $H$ would benefit the agent by providing more of the information contained in the state $s_t$. The only argument against large $H$ would be that it might necessitate larger model capacity required to represent $\pi_\theta$, risking overfitting to a finite demonstration dataset.

Yet, many prior works have reported [51, 26, 6, 11] that $H = 1$ is optimal. In other words, the *most poorly* observed setting, with $\tilde{o}_t = o_t$, yields the best results. Even more intriguingly, with $H > 1$, the likelihood $L(\theta^*)$ of held-out expert demonstrations improves, which means that there is no overfitting; only the environment reward $R(\theta^*)$ decreases. de Haan et al. [12] recently identified this problem as "causal confusion": BC policies misattribute expert actions to demonstration-specific nuisance correlates that no longer hold under the aforementioned distributional shift induced by policy execution.

We go one step further, pinning the nuisance correlate in a prominent class of causal confusion problems to the previous expert action $a_{t-1}^e$, which is often recoverable from observation histories, as in the car example in Sec 1. When expert actions are strongly correlated over time, the imitation learner has a suboptimal, but tantalizingly convenient shortcut [15]: merely learning to recover the

Table 1: MSE for next action prediction, conditioned on previous actions. The lower the error for a policy, the higher its tendency to generate actions that can be predicted from previous actions alone.

|  | Ant$\times 10^{-2}$ | Hopper$\times 10^{-3}$ | Humanoid$\times 10^{-1}$ | Reacher$\times 10^{-5}$ | Walker2d$\times 10^{-2}$ | HalfCheetah$\times 10^{-2}$ |
|---|---|---|---|---|---|---|
| expert | $6.91 \pm 0.21$ | $8.60 \pm 1.09$ | $6.93 \pm 0.32$ | $1.46 \pm 0.37$ | $2.47 \pm 0.07$ | $9.81 \pm 0.33$ |
| BC-OH | $0.66 \pm 0.04$ | $1.07 \pm 0.16$ | $0.18 \pm 0.01$ | $0.32 \pm 0.05$ | $0.46 \pm 0.02$ | $2.97 \pm 0.15$ |

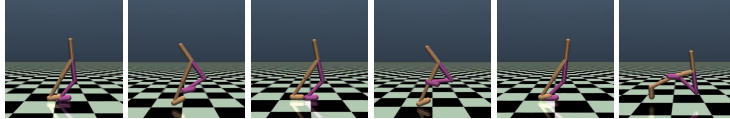

Figure 2: Frames of a copycat Walker2D agent falling (left to right): the right knee repeats previous actions at each time, failing to transition from an extended position to a bent position.

previous action very nearly maximizes the training objective $L(\theta)$. We call this the copycat problem, and posit that it accounts for many reported cases of causal confusion. Indeed, while de Haan et al. [12] propose to address the more general causal confusion problem, their experimental results are all in the copycat setting: they explicitly add $a_{t-1}$ into $o_t$ to induce causal confusion.

**Empirical Evidence for the Copycat Problem.** We now show qualitative and quantitative evidence demonstrating that the copycat problem occurs commonly in BC with partial observations. Motivated by robotic control, we use the six OpenAI Gym MuJoCo continuous control tasks. We set the history size $H = 2$ and train a neural network policy $\pi_\theta(a|\tilde{o}_t = [o_t, o_{t-1}])$ on expert demonstrations. We call this policy behavior cloning with observation histories (BC-OH). See Section 6 for more details.

To find a smoking gun for the copycat problem, we measure the predictability of the next action conditioned on the past actions, for a given policy. If the learner suffered from the copycat problem, this prediction would be easier on trajectories from the learned policy than on those from the expert policy. For each policy, we train a two-layer MLP to predict $a_t$ given $a_{t-1}, \ldots, a_{t-k}$ as input. Here we use $k = 9$. Table 1 shows the mean-squared error (MSE) on held-out trajectories, on all six MuJoCo environments. In each case, the MSE is lower for the cloned BC-OH policy, than for the expert, pointing to the copycat problem.

Figure 2 shows six frames of a copycat BC-OH agent falling in Walker2d, where the agent fails to switch its right knee from an extended to a bent position to maintain balance. Other cases are even worse: for example, we observed copycat Half-Cheetah agents that did not ever *start* to move from a resting position.

It is clear that BC-OH policies exhibit an increased tendency to repeat actions (or more precisely, to generate actions that can be predicted from previous actions alone). To what extent does this affect their performance? For each environment, we now measure: (i) the action predictability ratio, the log-ratio of expert and BC-OH errors in Table 1, and (ii) the normalized

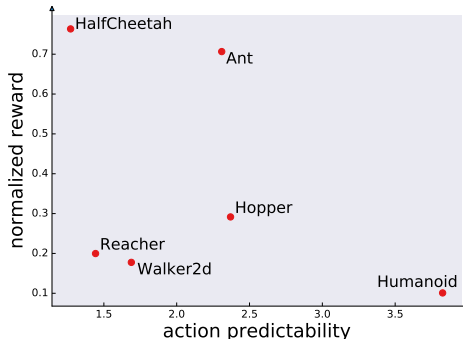

Figure 3: The action predictability metric (x-axis) versus the normalized reward (y-axis).

reward, the average BC-OH reward divided by the average expert reward. Figure 3 shows a scatter plot of these metrics, showing that the action predictability ratio is inversely correlated with the normalized reward. In other words, a copycat imitation policy that copies past actions rather than responding to observations tends to perform worse.

## 5 An Adversarial Solution to the Copycat Problem

We now propose an adversarial method to resolve the copycat problem. Our method builds on the standard behavioral cloning (BC) setup described in Sec 3. BC trains a policy $\pi_\theta(a_t|\tilde{o}_t)$ on expert demonstrations, to map from observation histories $\tilde{o}_t = [o_t, o_{t-1}, \cdots, o_{t-H+1}]$ to expert actions.

For notational simplicity, we now restrict ourselves to deterministic policies, so that $a_t = \pi_\theta(\tilde{o}_t)$. For a policy represented by a multi-layer neural network, $\pi_\theta$ is easy to write as a composition of two learned functions, an encoder $E$ and a decoder $F$: $a_t = \pi_\theta(\tilde{o}_t) = F(E(\tilde{o}_t))$. The output of the encoder network is a feature embedding $\boldsymbol{e_t} = E(\tilde{o}_t)$.

**Adversarial Nuisance Variable Prediction.** Fundamentally, the copycat problem arises from the fact that $\boldsymbol{e_t}$ contains information about the nuisance correlate $a_{t-1}$, and $F$ learns to rely heavily on this information to predict $a_t$. This might suggest the following strategy: remove all information about $a_{t-1}$ from the embedding $\boldsymbol{e}_t$, so that $F$ cannot rely on $a_{t-1}$ at all. In other words, we would train the encoder $E$ to maximize the conditional entropy

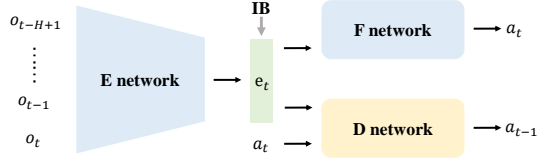

Figure 4: The network architecture. See Section 5 for explainations.

$H(a_{t-1}|\boldsymbol{e}_t)$, of the previous action, conditioned on the feature embedding. In practice, this means training an adversarial network $D$ to predict $a_{t-1}$ from $\boldsymbol{e}_t$.

However, note that removing all information about $a_{t-1}$ may be counterproductive: after all, the copycat problem arises only when $a_t$ and $a_{t-1}$ are highly correlated. Removing all information about $a_{t-1}$ would make it very difficult to predict $a_t$. Put a different way, in order to predict $a_t$ well, $F$ requires some information about $a_{t-1}$ to be retained in $\boldsymbol{e}_t$.

**Target-Conditioned Adversary (TCA).** To account for this, we design a slightly different adversarial strategy. Rather than removing all information about $a_{t-1}$ from $\boldsymbol{e}_t$, we would like to remove only the information about $a_{t-1}$ that is *not shared with* $a_t$. How might we set up an adversarial optimization process that removes this unshared information about $a_{t-1}$, while having no incentive to remove information that is actually useful for predicting the target variable $a_t$? The solution is simple: the adversary $D$ still tries to predict $a_{t-1}$ from $\boldsymbol{e}_t$, but with the target $a_t$ as an additional conditioning input. The resulting optimization would train the encoder $E$ to maximize the conditional entropy $H(a_{t-1}|\boldsymbol{e}_t, a_t)$.

Intuitively, this means that the optimization process has no incentive to strip $\boldsymbol{e}_t$ of any information about $a_{t-1}$ that is present in $a_t$ — removing that information in $\boldsymbol{e}_t$ would not affect $D$'s ability to predict $a_{t-1}$, since $a_t$ contains the same information. Thus, no information about the target variable $a_t$ is forcibly removed from the embedding. This means that the copycat solution of predicting the previous action is no longer a viable and convenient shortcut.

**Information Bottleneck (IB).** Further, we add an information bottleneck (IB) [1] between the encoder $E$ and the decoder $F$ to express the prior that any residual excess information in $\boldsymbol{e}_t$ should be ignored. Conceptually, our approach is built around identifying observation histories as likely to contain nuisance information. The target-conditioned Adversarial module only removes the information about $a_{t-1}$ and the IB further removes other nuisance information. IB has been used in other works in similar ways Alemi et al. [1], Rakelly et al. [37], Pacelli and Majumdar [31]. Specifically, we modify $E$ to predict the parameters $\boldsymbol{\mu_{e_t}}$ and $\boldsymbol{\sigma_{e_t}}$ of an independent normal distribution, from which $\boldsymbol{e}_t \sim p_E(\boldsymbol{e}_t) = \mathcal{N}(\boldsymbol{\mu_{e_t}}, \boldsymbol{\sigma_{e_t}})$ is sampled. To apply an information bottleneck, we penalize the KL divergence between this distribution and the unit normal $\mathcal{N}(\mathbf{0}, \mathbf{I})$. Finally, $D$ operates directly on $\boldsymbol{\mu_{e_t}}$, but $F$ sees the "noisy" sample $\boldsymbol{e}_t$. Intuitively, the IB implements a penalty for every bit of information that $E$ transmits to $F$, encouraging it to transmit only the most essential information and ignore the nuisance correlate.

Finally, putting the target-conditioned adversary and the information bottleneck together, we have the following min-max optimization problem:

$$\min_{E,F} \max_{D} V(E, F, D) = \mathbb{E}_{\boldsymbol{e_t} \sim p_E} \mathcal{L}(F(\boldsymbol{e_t}), a_t) + \lambda KL(p_E(\boldsymbol{e_t}) || \mathcal{N}(\mathbf{0}, \mathbf{I})) - \alpha \mathcal{L}(D(\boldsymbol{\mu_{e_t}}, a_t), a_{t-1}),$$

where $\mathcal{L}$ is an appropriate regression loss, such as the mean squared error.

**Implementation Details.** In our implementation, $E$, $F$, and $D$ are all represented by neural networks. The overall network structure is depicted schematically in Fig 4. We train with stochastic gradient descent using the Adam optimizer. We use the reparametrization trick [24, 38] to evaluate the gradient through the expectation in the first term. We found that (1) different learning rates for $E$, $F$ and $D$ and (2) noise on the embedding $\boldsymbol{e}_t$ when training $D$, are helpful during optimization. In

our experiments, the encoder $E$ is a 4-layer MLP, and the decoder $F$ and the adversary $D$ are each 2-layer MLPs. More details and pseudocode are in Appendix.

**Connection to de Haan et al. [12].**    As mentioned above, the phenomenon of causal confusion in imitation learning was identified in [12]. Their method, which we call CCIL, targets causal confusion in three steps: disentangled representation learning, policy learning conditioned on exponentially many causal graph structures, and online targeted interventions through environment interactions. They demonstrate results on simplified environments where a nuisance correlate is explicitly added into the state. In comparison, we focus on a more specific, but widely prevalent form of causal confusion (see Section 4), where the previous action is the nuisance correlate in imitation learning from observation histories — the copycat problem. This knowledge of the nuisance correlate drives our simpler, more robust, and more scalable approach, and allows us to perform experiments in more realistic settings: partially observed MDPs with history-aware imitation learners. Further, unlike CCIL, we demonstrate results on purely offline imitation: the imitation learner does not ever need to access the environment during training. We show experimental comparisons in Sec 6.

# 6    Experiments

In this section, we conduct experiments to evaluate our method against a variety of baselines. We also qualitatively study our method in order to better understand the newly introduced algorithm.

## 6.1    Baselines

We compare our method to the following baselines.

**Behavioral cloning (BC-SO, BC-OH and RNN).**    **BC-SO** is naive behavioral cloning (Sec 3) with $H = 1$, which does not suffer from the copycat problem, since it cannot infer the previous action. However, BC-SO might suffer from lacking information to make an action decision. **BC-OH** is naive BC with $H > 1$. It allows the agent to access more of the state information necessary for optimal action selection, but it is prone to the copycat problem. We set $k$ to 2 in our experiments. We train BC-OH agents both with stacked inputs to a feedforward policy, and with sequential inputs to an RNN policy.

**Dropout-BC [6].**    To combat causal confusion, Bansal et al. [6] add a neural net dropout layer [45], to the subset of inputs that might causally confuse the agent. Like our method, this method also assumes knowledge of the nuisance correlate. For this baseline, we add a dropout layer on the historical observations, i.e. $o_{t-1}, o_{t-2}, \cdots$.

**CCIL [12].**    This is the method proposed in de Haan et al. [12], discussed in Sec 5. Note that our method assumes offline imitation, where we only have access to a pre-collected dataset, without any in-environment interactions. CCIL requires access to both. We set the number of environment interactions to 100.

**DAGGER [39].**    DAGGER is a widely used method to correct distributional shift in imitation learning. Similar to CCIL, DAGGER also requires environment interaction, with access to a queryable expert. We evaluate DAGGER at two values for the number of environment queries: 100 and 1000.

**RL Expert [41].**    We also compare against the RL expert, trained with TRPO [41], that was used to generate the expert data for imitation.

Aside from these baselines, we evaluate several ablations of our approach: Ours w/o Adversary, Ours w/o TCA, and Ours w/o IB, corresponding respectively to omitting the $D$ network entirely, omitting only the target-conditioning in $D$, and omitting the information bottleneck.

## 6.2    Partially Observed Environments

Motivated by robotics applications, we evaluate our approach on all six MuJoCo [48] control environments from Open AI Gym: Ant, HalfCheetah, Hopper, Humanoid, Reacher and Walker2D. These tasks vary broadly in their state and action spaces, environmental dynamics, and reward structure. To evaluate the copycat problem, we make these environments partially observed in a natural way by setting the states to only include the joint positions and exclude other information i.e. velocity and external force. We train TRPO [41] agents without partial observability to generate

Table 2: Cumulative rewards per episode in partially observed (PO) environments. The top half of the table shows results in our offline imitation setting. The lower half shows methods that additionally interact with the environment, including accessing reinforcement learning rewards and queryable experts. CCIL cannot run on Ant and Humanoid because of their high-dimensional observations.

| | PO-Ant | PO-Hopper | PO-Humanoid | PO-Reacher | PO-Walker2d | PO-HalfCheetah |
|---|---|---|---|---|---|---|
| BC-SO | 1300 ±148 | 275 ±40 | 587 ±58 | −79 ±5 | 363 ±86 | −38 ±36 |
| BC-OH | 1750 ±146 | 293 ±83 | 565 ± 80 | −64 ±4 | 592 ±124 | 820 ±60 |
| BC-OH (RNN) | −311 ±150 | 315 ±32 | 367 ± 64 | −75 ±5 | 190 ±14 | 830 ±398 |
| Dropout-BC [6] | 830 ±330 | 223 ±49 | 577 ±65 | −80 ± 7 | 283 ±194 | 406 ±165 |
| Ours w/o Adversary | 2030 ±88 | 473 ±129 | 638 ±101 | −70 ±3 | 962 ±189 | **1260** ±68 |
| Ours w/o TCA | 1629 ±287 | 322 ±74 | 607 ±58 | −55 ±3 | **1310** ±333 | 795 ±398 |
| Ours w/o IB | 1970 ±107 | 683 ±132 | **696** ±48 | −57 ±4 | 929 ±266 | **1260** ±44 |
| Ours | **2150** ±34 | **1086** ±262 | 671 ±61 | **−54** ±4 | 1296 ±288 | 1250 ±42 |
| RL Expert [41] | 2348 ±5 | 1780 ±0 | 4963 ±47 | −10 ±0 | 2428 ±2 | 1336 ±4 |
| CCIL [12] | - | 145 ±55 | - | −68 ±6 | 474 ±134 | 714 ±132 |
| DAGGER [39] (100 queries) | 2090 ±34 | 978 ±216 | 688* ±47 | −52 ±4 | 701 ±133 | 1080 ± 86 |
| DAGGER [39] (1k queries) | 2240 ±14 | 1120 ±203 | 812* ±99 | −15 ±2 | 2170 ±174 | 1270 ± 22 |

\* for Humanoid, we used 100k and 500k queries for DAGGER, instead of 100 and 1000.

expert demonstrations. See appendix for more details of those environments as well as the collection of the demonstration dataset.

### 6.3 Results and Analysis

**Question 1.** *Does our method improve performance over baseline approaches for behavioral cloning from observation histories?*

For each method, in each environment, we train five policies with varying random initializations, and report the mean and standard deviation of their cumulative rewards. Table 2 shows these results. Our full method performs best, or tied best across all six environments, among purely offline imitation methods. As expected, behavior cloning from single observation (BC-SO) performs poorly, due to partial observability. BC-OH, with observation histories, helps to varying extents in five out of six environments, but still performs much worse than the RL expert that generated the imitation trajectories. While the RNN hidden state can be thought of as implementing a natural information bottleneck, we find it performs similarly or worse than the feedforward BC-OH policies. Dropout-BC [6] was originally proposed and evaluated in a setting where the nuisance correlate corresponded to a single dimension in the input. This is not true in our settings, where the nuisance variable is a function of the high-dimensional past observations. It performs uniformly poorly across all tasks.

Among non-offline imitators, CCIL fails to achieve better rewards than our approach, even with additional environmental interaction and queryable experts. We believe that this is because it relies on learning a disentangled representation to handle high-dimensional observations, which it was unable to effectively do in our environments. DAGGER (100 queries) performs comparably with our purely offline method, and DAGGER (1000 queries) performs significantly better, approaching the performance of the RL expert in several settings. On Hopper and Humanoid, even the best imitators fall far short of the expert.

**Question 2.** *Which components of our method are most important to its performance?*

Comparing ablated variants of our approach in Table 2, the target-conditioned adversary (TCA) and the information bottleneck (IB) are both clearly important components. Our method without TCA performs the poorest out of our ablations in four out of six environments — recall that the unconditioned adversary could force the removal of too much important information from the learned embedding. Using only the information bottleneck without an adversary (Ours w/o Adversary) already yields significant improvements over BC-OH and Dropout-BC — we take this to mean that merely penalizing the description length of the learned representation encourages dropping nuisance information. Ours w/o IB, which only uses TCA, does nearly as well as our full method on most tasks, suggesting that the target-conditioned adversary is the most important component of our approach.

**Question 3.** *Does our policy truly rely less on the previous action $a_{t-1}$?*

We train an MLP, in Hopper, to predict $a_{t-1}$ from $[e_t, a_t]$ — recall that this is the function of the $D$-network in our algorithm. We use the test mean-squared error (MSE) score of this trained predictor as a measure of the confounding excess information: the higher the score, the less excess information

about $a_{t-1}$ there is in $e_t$, and the better for avoiding copycat issues. For comparison, we compute the same MSE score for the BC-OH baseline. Finally, as an upper bound, we compute the MSE score for predicting $a_{t-1}$ from $a_t$ alone. Table 3 shows these scores. As expected, our approach w/o IB (i.e. TCA alone) removes nearly all of the confounding information from $e_t$, making $a_{t-1}$ no easier to predict than from $a_t$ alone. In other words, our approach successfully trains policies that do not overly rely on the previous action.

Table 3: The mean squared error for predicting the previous action $a_{t-1}$ from various features.

| setting | w/o IB | BC-OH | $a_t$ |
|---|---|---|---|
| test MSE $\times 10^{-3}$ | $\mathbf{5.97} \pm 1.32$ | $0.38 \pm 0.04$ | $5.00 \pm 0.33$ |

**Question 4.** *Does the TCA help to retain information that is useful for predicting the next action?*

Table 4 reports the BC testing error (next action prediction MSE) in Hopper for 3 methods: BC-OH, ours w/o IB (i.e. TCA only), and ours w/o IB and TCA (i.e. unconditional adversary loss only). While TCA cannot predict current action $a_t$ as well as BC-OH, its performance is significantly better than the unconditional adversarial setting, indicating that the target-conditioning effectively preserves more information about the next action. Note that in terms of actual environmental reward, TCA performs the best, followed by the unconditioned adversary.

Table 4: The BC mean squared error (for predicting the next action $a_t$) on test data.

| setting | w/o IB | w/o IB&TCA | BC-OH |
|---|---|---|---|
| test MSE $\times 10^{-3}$ | $9.48 \pm 1.89$ | $14.62 \pm 1.41$ | $\mathbf{0.89 \pm 0.12}$ |

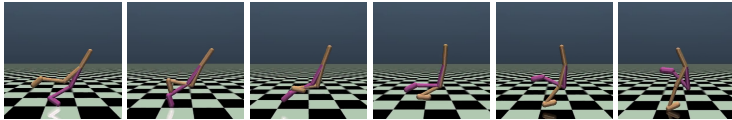

Figure 5: Our method solves the copycat problem, demonstrated in Walker2D. The agent no longer only moves one leg, but it starts to walk with both legs, in contrast to the BC-OH agent in Figure 2.

**Question 5.** *Does our policy repeat itself during test time?*

In Section 4, we show that the BC-OH baseline tends to repeat its own previous action during online rollouts. Here we use the same action predictability metric to test our method. Table 5 shows the result. We can see that our method reduces the correlation between consecutive actions. Figure 6 shows the impact on performance: our method reduces action predictability and increases reward in every environment, indicating that it successfully addresses the copycat problem.

Besides the action predictability score in Table 5, we now present an example of behavior where the copycat problem is visibly resolved. To compare with Figure 2, we visualize the Walker2D agent again in Figure 5. Our agent responds to its observations, rather than copying the previous action as BC-OH often did in Figure 2. The result is a doubling of the mean reward in this environment from BC-OH's 592, to our 1296 (see Table 2).

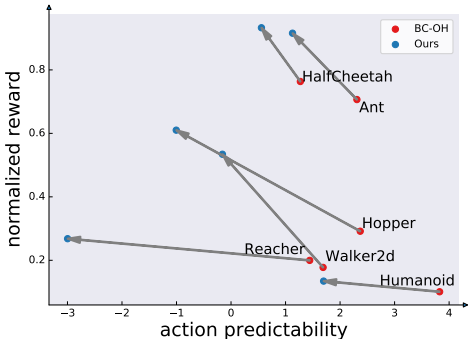

Figure 6: In all 6 environments, our method reduces action predictability, and improves the normalized reward, over the BC-OH baseline.

Table 5: Similar to Table 1, the predictability of the next action conditioned on past actions of our method and the BC-OH policy. Our method reduces the repetition of actions over time.

| | Ant$\times10^{-2}$ | Hopper$\times10^{-3}$ | Humanoid$\times10^{-1}$ | Reacher$\times10^{-5}$ | Walker2d$\times10^{-2}$ | HalfCheetah$\times10^{-2}$ |
|---|---|---|---|---|---|---|
| BC-OH | $0.66 \pm 0.04$ | $1.07 \pm 0.16$ | $0.18 \pm 0.01$ | $0.32 \pm 0.05$ | $0.46 \pm 0.02$ | $2.97 \pm 0.15$ |
| Ours | $2.20 \pm 0.06$ | $2.26 \pm 0.12$ | $1.33 \pm 0.02$ | $2.99 \pm 0.36$ | $3.07 \pm 0.08$ | $5.40 \pm 0.20$ |

# 7 Conclusion and Future Work

In this paper, we identify the copycat problem that commonly afflicts imitation policies learning from histories of observations. We systematically study this phenomenon by carefully designing a set of diagnostic experiments, which shows the existence of this problem in multiple environments. Finally, we propose a new adversarial mechanism to tackle this problem. We validate our approach through extensive comparisons with previous approaches on 6 standard continuous control benchmark tasks. Our method significantly alleviates the copycat problem for offline behavioral cloning from observation histories, and even outperforms some existing online behavioral cloning methods that have additional access to the environment. As for future work, while our method works pretty well in state-based environments, the improvement in image-based experiments is not significant yet. The causal confusion in image-based control is still an open question and we hope to address these more realistic scenarios in the future.

# 8 Broader Impact

In this paper, we introduce a systematic approach to combat the "copycat" problem in behavioral cloning with observation histories. Behavioral cloning can be applied to a wide range of applications, such as robotics, natural language, decision making, as well as economics. Our method is particularly useful for offline behavioral cloning with partially observed states.

Offline imitation is currently one of the most promising ways to achieve learned control in the world. Our method can improve the real world performance of behavior cloning agents, which could enable wider use of behavior cloning agents in practice. This could help to automate repetitive processes previously requiring human workers. While on the one hand, this has the ability to free up human time and creativity for more rewarding tasks, it also raises the concerning possibility of the loss of blue collar jobs. To mitigate the risks, it is important to promote policy and legislation to protect the interests of the workers who might be affected during the adoption of such technology.

# 9 Acknowledgements

We are grateful to Pim de Haan, Jianing Qian, and Sergey Levine for fruitful discussions and comments.

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
