[Supplementary Material]

## 10 Appendix

### 10.1 Capability of each method to resolve the copycat problem

Figure 7: Normalized reward scores vs. action predictability. Arrows for each method start at BC-OH performance and end at the method's performance.

In Figure 6, we showed that our method decreases action predictability, i.e. reduces the action repeat, and increases reward across all environments, over BC-OH baseline. Here, we present this result environment-wise, with additional arrows corresponding to baselines and ablations. The red arrow corresponds to our full method.

In nearly all environments, our approach yields the highest reward of all offline methods, and also achieves the lowest action predictability score. In three out of six environments, it approximately matches the performance of the even the best online imitation approach (DAGGER 1000, cyan).

### 10.2 Network architecture

Figure 8: The architecture of target-conditioned adversarial model.

The network architecture of target-conditioned adversarial model is shown in Figure 8. We update the parameters in the neural network by the process shown in Algorithm 1.

---

**Algorithm 1** Minibatch stochastic gradient descent training of the objective function.

---

**Require:** learning rate schedule of $E$ and $F$, $l_{EF}$; learning rate schedule of $D$, $l_D$; embedding noise std $\sigma$; batch size $m$; number of frames in the imitation learning $H$

  **for** number of training iterations **do**
- Sample minibatch of $m$ examples $\{\boldsymbol{S}^{(1)}, \ldots, \boldsymbol{S}^{(m)}\}$, where each
$\boldsymbol{S}^{(i)} = \{(o_{t_i-H+1}, a_{t_i-H+1}), \ldots, (o_{t_i}, a_{t_i})\}$ is a stack of $H$ frames in the expert demonstration
- Update $\theta_E$ and $\theta_F$ by *descending* its gradient $\partial V/\partial \theta_E$ and $\partial V/\partial \theta_F$, with learning rate $l_{EF}$.
- Generate minibatch of $m$ noise samples $\{\boldsymbol{\epsilon}^{(1)}, \ldots, \boldsymbol{\epsilon}^{(m)}\}$, where $\boldsymbol{\epsilon}^{(i)} \sim \mathcal{N}(0, \sigma^2 \boldsymbol{I})$.
- Add noise $\boldsymbol{\epsilon}^{(i)}$ to $\boldsymbol{e}^{(i)}$ in the computation graph, i.e. $\boldsymbol{e}^{(i)} = \boldsymbol{e}^{(i)} + \boldsymbol{\epsilon}^{(i)}$ [3, 43]
- Update $\theta_D$ by *ascending* its gradient $\partial V/\partial \theta_D$, with learning rate $l_D$.

  **end for**

---

Figure 9: The architecture of information bottleneck model. The E network is pretrained by the target-conditioned adversarial model.

Our Information Bottleneck part, as shown in Figure 9, uses the encoder $E$ pre-trained by the target-conditioned adversarial (TCA) model to initialize its encoder $E$. We add a information bottleneck module to the embedding generated by its encoder $E$ and denote the combination of the bottleneck and the original $F$ network as $IB$-$F$, which is optimized by the supervised loss $Loss(\hat{a_t}, a_t)$ and $KL(e_t, N(0, 1))$.

## 10.3 Experiment Environments

Motivated by robotics applications, we compare our method with baselines and ablations on all six MuJoCo [48] control environments from Open AI Gym. Snapshots of these six environments are shown in Fig 10. The goal of Ant, Humanoid, Walker2d and HalfCheetah is to make the agent walk or run as fast as possible while Hopper's objective is to make the agent hop forward as fast as possible and Reacher's objective is to enable the robot reach a randomly located target.

## 10.4 Expert Data Collection

To collect expert demonstrations, we first train an expert with reinforcement learning, specifically TRPO [41]. This expert policy is executed in the environment to collect demonstrations. Since the six environments have very different observation dimensions, we collect 1k transitions for HalfCheetah, Reacher and Ant, 20k transitions for Hopper and Walker2D, and 200k transitions for Humanoid. The demonstration set size is roughly linear to the number of observation dimensions.

## 10.5 Implementation details

### 10.5.1 Experiment setting

We use the OpenAI gym package to construct the experiment environments and set the frame skip in all environments to 1. We use Adam optimizer and mean squared error as our loss function. During

Figure 10: Snapshots of the six MuJoCo environments used in our experiments, including Ant, Hopper, Humanoid, Reacher, Walker2d and HalfCheetah (from left to right).

training, we set the number of training iterations to 300,000 and minibatch size to 64. And we decay the learning rate by 0.1 three times during the course of training.

We evaluate each model in environments at five checkpoints (280k, 285k, 290k, 295k and 300k iteration) and calculate their average rewards as the final reward.

### 10.5.2 Hyper-parameters

We set the learning rate of $E$ and $F$ network to $2 \times 10^{-4}$, the weight of adversarial loss $\alpha$ to 2 and the weight of KL divergence $\lambda$ to $1 \times 10^{-3}$ in all the environments. The hyper-parameters that differ among different environments are shown in Table 6.

Table 6: Hyper-parameters used in each environment. The embedding noise is used in all environments except Hopper and Walker2d to stablize training.

|  | D learning rate | D embedding noise std |
|---|---|---|
| Ant | $5 \times 10^{-4}$ | 2.0 |
| Hopper | $4 \times 10^{-4}$ | - |
| Humanoid | $4 \times 10^{-4}$ | 1.5 |
| Reacher | $4 \times 10^{-4}$ | 2.0 |
| Walker2d | $2 \times 10^{-4}$ | - |
| HalfCheetah | $4 \times 10^{-4}$ | 2.0 |

### 10.6 Test Loss Comparison

As shown in Table 7, the BC-OH baseline has smaller test loss than BC-SO, indicating that there is no over-fitting in training with observation histories. Thus the performance drop from BC-SO to BC-OH, such as in Humanoid, is attributed to the copycat problem. And compared with BC-OH, our method has higher test loss while producing higher reward when evaluated in the environment, showing that we're actually solving the copycat problem rather than only reducing overfitting to training data.

Table 7: The test loss of BC-SO, BC-OH and our method under the same settings as Table 2.

|  | Ant | Hopper | Humanoid | Reacher | Walker2d | HalfCheetah |
|---|---|---|---|---|---|---|
| BC-SO | 0.1079 | 0.0077 | 0.5710 | 0.0008 | 0.0410 | 0.1225 |
| BC-OH | 0.0829 | 0.0049 | 0.5470 | 0.0007 | 0.0136 | 0.0309 |
| Ours | 0.0860 | 0.0087 | 0.5632 | 0.0010 | 0.0229 | 0.0321 |