[Reviews · NeurIPS 2020]

Review 1

Summary and Contributions: This paper first identifies the "copycat" issue on the Behavioral Cloning problem by showing both quantitative and qualitative experiments. The authors then propose an adversary method by training another adversarial network to encourage the representation used by the policy network to ignore the information not shared by the next step action. The authors then evaluate the proposed method from multiple aspects with experiments and analysis.

Strengths: - The authors identify an important issue in Behavior Cloning with offline expert demonstration data named copycat problem in the paper, or casual confusion in previous works. The paper does a good job of motivating the problem existed in Behavioral Cloning in the introduction. I also enjoy reading the section where the authors demonstrate the existence of this problem in a clean way first before directly diving into the solution. - The paper is well-written with good illustrations in both textual and visual format. Overall, I think the presentation of the paper is in good shape. - Experiments conducted by the authors carefully inspect the effectiveness of the approach. This is important to validate the approach. In addition, the authors did a good job on comparisons and necessary discussions with prior works.

Weaknesses: - Although I enjoy reading the paper overall, I do have some major concerns with the experiments in this paper. First, the authors trained the Mujoco agents with TRPO as expert demonstrations. Although I understand that this is a common way for Mujoco agents to acquire demonstrations, this method indeed has some drawbacks that do not cover real-world demonstration scenarios. For example, real-world demonstrations may not be always optimal and very noisy. None of these aspects can be covered with the current Mujoco experiments. This leads to my second concern below. - The experimental setup is too constrained. Even though the paper heavily discussed and compared with CCIL and DAGGER, the experiments are in very different domains and the setups are far less complex than the original two works. For example, CCIL covers experiments in self-driving cars and Atari games which is both more complex and includes more stochastic factors. However, in the current setting, the environment is deterministic and the expert data is very "clean". The TRPO also observers the full state of the environment to generate the demonstration data, which further validates my above point. - Thus, although I like the thorough analysis presented by the paper, the experiments are far less complex thus make the effectiveness of the algorithm in complex domains susceptible. I strongly encourage the authors to consider similar experiments in previous works. - Please add error bars to the MSE results for action predictability across the paper. Minor: - typo at line 46: should be "poorer"

Correctness: Yes, although there are some major concerns regarding the experiments as mentioned above.

Clarity: Yes.

Relation to Prior Work: Yes.

Reproducibility: Yes

Additional Feedback: =============================== Post rebuttal: I think the authors did a good job during rebuttal to resolve my concerns on the task / environment complexity and stochasticiy. I thus increase my score to recommend acceptance.


Review 2

Summary and Contributions: The paper proposes to tackle a specific variant of the causal confusion problem (Haan et al., 2019) in imitation learning, namely the "copycat" problem, where a history-dependent BC agent learns to predict a_t by coping the previous action a_{t-1}. This phenomenon is prevalent in many continuous control problem. The paper proposes a simple solution that alleviate this problem in an offline imitation learning setting and is evaluated on a number of standard continuous-control benchmark tasks.

Strengths: + The causal confusion problem is a new and important problem to study, especially in a pure offline setting. Haan et al. formalizes the problem but provides a solution that requires online intervention. Although the method proposed by this paper is only for a specific variant of the causal confusion problem, the method is conceptually simple and works in a pure offline setting. + The method is described with great conciseness and clarity

Weaknesses: - The key claim made by this paper is that history-dependent BC policy (BC-MO) suffers from the copycat problem because the history contains enough information about the past action taken by the policy. However, the motivating example presented in Table 1 and line 132-145 is not really a convincing evidence for the copycat problem. An alternative explanation for the result is that the BC policy is known to suffer from the mode collapse problem, where if there are contradicting supervisions given the same state due to noises, BC learner tend to "average out" the noise, making the action more predictable. A more convincing example would be to show that the actions from a history-independent BC policy (BC-SO) is more difficult to predict than that of the BC-MO policy. - Another implicit assumption made by the paper is that learning to infer past actions (a_{t-N}, N > 0) from observation histories are easier than learning to predict the action directly from the observations. This claim might be true for most environments, but would there be cases where this is not true? I'd like to see a conceptual analysis of this assumption in the paper. - The Information Bottleneck (IB) seem ad-hoc and orthogonal to the main claim of the paper. The intuition behind IB provided by the paper "Intuitively, the IB implements a penalty for every bit that E transmits to F, encouraging it to transmit only the most essential information" is a bit hand-wavy and it'd be great to see some theoretical justifications to this claim.

Correctness: Seems to be correct.

Clarity: The paper is well-written an easy to follow.

Relation to Prior Work: Yes.

Reproducibility: Yes

Additional Feedback: ======================================= AFTER REBUTTAL: I'm satisfied with the author response. Thus I uphold my original assessment and recommend accepting the paper.


Review 3

Summary and Contributions: The authors discuss a failure mode of imitation learning algorithms in a very specific setting, in which we provide the imitation policy with a backward-looking history of observations as input, rather than a single observation. The authors hypothesize that windowed observations can be used as a “simple fix” for contending with partial observability ------ after rebuttal ------- The authors answered a number of my questions and included new results that I requested. Given this rebuttal, I'm increasing my score. The authors then name a hypothesized failure mode the “copycat problem” — in which the IL policy falls into an easy-to-find local optimum in which learns to repeat past low-loss actions at key transition states (incurring a small penalty at these relatively-rare transitions), rather than learning a more globally-optimal policy which correctly disentangles “nuisance correlates” in the observation/action data to produce a correct action at the transition. The remainder of the paper is essentially in two parts, which constitute the bulk of the proposed contributions: (1) the authors perform a thorough empirical analysis to prove the existence of this problem on robotics simulation domains, and then (2) propose an adversarial policy architecture loss designed to bottleneck the amount of information on past expert actions which is identifiable from the policy network input, followed by an experimental evaluation to other methods on simulation domains, showing their method outperforms previous methods in those domains.

Strengths: This paper is very well-executed methodologically and stylistically — the explanation of the problem setting, empirical investigation of the hypothesized root cause (the “copycat problem”), exposition of the proposed method, and its evaluation are all crisp, easy-to-understand, well-analyzed. I enjoyed reading it and felt I understood the material well without deciphering dense text or repeatedly referring back to previous papers for notation and jargon. The method proposed is a reasonably-novel solution to a reasonably-relevant problem mentioned in some of the citations, especially in autonomous driving applications. However, I have concerns about the paper’s ability to justify the method’s significance and relevance to the field as a whole (and important application domains), which I discuss below.

Weaknesses: This paper’s main weaknesses revolve around the author’s justification (or lack thereof) of the problem setting and proposed solution method in the context of other possible solutions to the original problem (partial observability) rather than the implementation problem discussed (the “copycat” problem). Specifically, conditioning policies on causal sequences is a well-known tactic for contending with partial-observable MDPs (POMDPs) (See RL^2 (Duan et al 2017), Schmidhuber, etc), but the authors do not mention, and do not evaluate, policy architectures which are explicitly designed for sequence modeling (e.g. RNNs or autoregressive attention models). If the “simple fix” for partial observability (Line 23-24) is sequence-conditioned policies, but conditioning feedforward policies on sequences leads to a failure mode which has its roots in the policy’s inability to model the passage of time (the “copycat problem”), why did the authors not at least consider for the reader the possibility of switching to these well-known sequence modeling architectures? In the original problem context, where we have a system which works well with MDPs (feedforward policies and BC) which we would like to now use for POMDPs, conditioning on causal windows is indeed a “simple fix.” However, the method (a “fix” for the “simple fix”) proposed here (a complex adversarial policy architecture and associated loss) is far from simple. It is at least as complex as switching to RNNs and autoregressive models, and so warrants a fair comparison to those well-known options. I worry that this paper may be “missing the forest for the trees” for the reader. I would encourage the authors to give a reasonably-comprehensive context for the paper's contents. This paper presumes that the reader has already decided to limit themselves to a very narrow model class (feedforward), then beautifully shows how to overcome a major downside of that model class in a fairly complex way. It never justifies (or even mentions, really), why the reader should limit themselves to such a restrictive set of policy models in the first place, it doesn’t compare that choice to other obvious ones (e.g. using RNNs), and it doesn’t allow the reader to balance the pros and cons of that decision by showing how its method compares to the original BC loss, but a model class better-suited to the problem.

Correctness: I believe the paper and empirical methodology are correct and the empirical experiments and comparisons appear to live up to a high standard.

Clarity: The paper is very well-written. I especially appreciated the authors’ clear exposition of the problem, easy-to-understand analysis, plots and figures, and well thought-out diagrams.

Relation to Prior Work: This is a major weakness of the paper. The crux of the method is an information bottleneck which prevents irrelevant past information from corrupting the current choice of actions. RNNs specifically create a natural information bottleneck (though not enforced with stochasticity) because the amount of information they can transmit through the hidden state is limited in practice. Other techniques for contending with partial observability in IL/RL are completely missing. The most proximate recent citation for this line of work would start with RL^2 (Duan et al 2017). See also: https://arxiv.org/pdf/1507.06527.pdf https://ieeexplore.ieee.org/abstract/document/4220824?casa_token=8bETkGG1J8AAAAAA:lPrMlU2EBY6XbJiKpc3wuuD5urfcouD593-_ftwkQY3X2cXlWEjF-BpxyzhTNrBXkeesidihRA

Reproducibility: Yes

Additional Feedback: Nits: Figure 3: This needs a line of best fit and an R^2 value. There seems to be some correlation (as claimed), but Reacher and Walker2d are outliers which need to be explained. Line 46 (grammar): “more poorer” —> worse


Review 4

Summary and Contributions: The paper analyzes the subproblem of causal confusion which is present when training agents via behavioral cloning. Specifically, the paper focuses on the copycat problem, i.e. the tendency of a policy to naively copy the previous expert action. As a solution, the authors propose to regularize the latent representations of the policy such that they do not contain information about the previous action to make the naive solution of coping the previous action impossible.

Strengths: The copycat problem is an important problem which greatly limits applicability of BC, especially in partially observable environments. Therefore, I find this paper relevant to the community. The solution is sound. The paper does not oversell their method. They sincerely explain that the problem which is addressed is the copycat problem alone, not BC causal confusion as a whole. I agree, however, that this subproblem is a widely prevalent form of causal and worth dedicated method. The proposed method is sound and the results for relevant ablations (along with important baselines) are presented. I enjoyed reading the additional analyses about MSE predictor which really makes the point. I wouldn’t say that the solution is very novel from algorithmic perspective (it builds on known components from domain adaptation), but it is intuitive, not overly complicated, rather easy to implement (similarly hard as GAIL implementation, so popular in imitation learning now) and seems to work. All we need :)

Weaknesses: The experiments are done only from low-dimensional states. The authors tried the solution for Walker2d from pixels (see supplementary material 9.7) but the difference between the proposed method and BC does not look to be significant (the std confidence intervals are not provided, although I assumed the same as for other methods for this task from Table 2). Also, even in this case, they do not train policy end-to-end from pixels (and I do not know why). The main point of the work is to apply BC in partially observable environments. The used environments are only artificially done partially observable (removing joint velocities) but I would prefer to see the method applied on environments designed to be POMDPs. I would like to see a comparison with CCIL when more environment interactions are allowed (similarly as you present two variants for DAGGER). Now, it is a bit suspicious that only 100 interactions are allowed, which makes me think that the CCIL method would be better than the proposed method which allowed to interact more. But that would be okay! Online interactions are expensive (especially on the real robot), and it is understandable that methods leveraging them work better. It would be nice to know the difference though.

Correctness: All good.

Clarity: Overall the clarity is really good. In Table 2, I would bold not only the best values for a task, but also all which are not significantly worse (or at least mark them somehow). Now, I had to manually compare the confidence intervals to check which methods are actually worse.

Relation to Prior Work: The authors acknowledge relevant prior work and directly compare with the most relevant ones in their experiments. I would recommend to mention the work by Geirhos et al. about shortcut learning. I think that BC causal confusion (as defined by de Haan et al.) and your copycat problem are special instances of shortcut learning. Additionally, there are works in adversarial imitation learning which also regularize the latent representation to account for shortcut learning related problems. For example, third person imitation learning (Stadie et al), variational discriminator bottleneck (Peng et al.) and task-relevant adversarial imitation learning (Zolna et al.). These works, however, focus on online settings and I do not find it necessary to compare with them in your experiments. So all good! :)

Reproducibility: Yes

Additional Feedback: AFTER REBUTTAL: I have read other reviews, which made me aware of a few weaknesses I missed. However, I think that the authors addresses the most important points and I would be reluctant to reject the paper. I kept my score.

[Author Response · NeurIPS 2020]

We would like to thank all the reviewers for taking time to read our paper and providing valuable suggestions. We're
happy to see that reviewers like our paper and we provide new experiments with explanations to address their concerns.

**R3: No comparison to RNN architectures which create natural bottlenecks.** Thank you for this suggestion. We
now replace the fully-connected history-stack encoder network in our BC-MO baseline with an RNN sequence encoder
of similar size and output dimension. The table below shows episode rewards for the original fully-connected BC-MO
vs. this new RNN variant. RNN scores are mostly on par with BC-MO, and much worse in Ant and Walker2D envs.

|  | PO-Ant | PO-Hopper | PO-Humanoid | PO-Reacher | PO-Walker2d | PO-HalfCheetah |
|---|---|---|---|---|---|---|
| BC-MO (Feed-forward) | 1750 ±146 | 293 ±83 | 565 ± 80 | −64 ±4 | 592 ±124 | 820 ±60 |
| BC-MO (RNN) | −311 ±150 | 315 ±32 | 367 ± 64 | −75 ±5 | 190 ±14 | 830 ±398 |

**R3: Other techniques contending with partial observability, missing citations.** Thank you for this suggestion;
we'll set up this broader context in introduction, and include a paragraph in related work.

**R1: Why TRPO as expert demonstrations? Expert data too "clean".** As R1 notes, using an RL agent as the expert
is standard in imitation learning, e.g. DAgger, GAIL, and CCIL, which is the most closely related prior work to ours.
We have now generated new noisy demonstrations by executing the trained RL expert in $\epsilon$-greedy exploration mode
($\epsilon = 10\%$), sampling exploration actions from $U[-1, 1]$. Although the expert as well as all imitators perform worse
now, our method still performs slightly better than BC-MO. For example, in Reacher, reward is $-68 \pm 3$ for BC-MO
vs. $-61 \pm 10$ for ours; in HalfCheetah, reward is $97 \pm 29$ for BC-MO, vs. $154 \pm 82$ for ours. We will include results
from all 6 environments in camera-ready.

**R1&R4: Results in high dimensional and naturally partially observed environments.** We have now performed
two new experiments on Atari Enduro and UpNDown. As in CCIL, we use a $\beta$-VAE to encode images. We use 10k
transitions for Enduro and 200k transitions for UpNDown. Our method outperforms BC-MO and CCIL.
In Enduro, the rewards are **Ours**: $27 \pm 1$, BC-MO: $24 \pm 4$, CCIL: $13 \pm 2$ . Expert: $52 \pm 1$.
In UpNDown, the rewards are **Ours**: $54 \pm 3$, BC-MO: $50 \pm 2$, CCIL: $26 \pm 6$. Expert: $64 \pm 4$.
Note: the original CCIL paper experiments don't use observation histories, and CCIL struggles with these higher dimensional inputs.

**R4: CCIL with more interactions?** Compared to our approach, CCIL requires *additional* environmental interactions
aside from the demonstration data. In the paper, we use a comparable number of interactions to the original CCIL paper.
We have now increased the number of interactions from 100 to 1000 for Hopper, improving the reward from 144 to 224,
but still much poorer than ours (1086). With even more interactions and a well-disentangled representation, CCIL may
be able to eventually outperform ours, but as R4 points out, that would not undermine our purely offline approach.

**R2: Why easier to infer past actions than the next action? Always true?** Indeed, this is not always true, and we did
find environments where behavior cloning from observation histories did not manifest the copycat problem, e.g., Atari
Pong. More broadly, inferring past actions is an example of a "shortcut", as R4 points out. As Geirhos et al, "Shortcut
Learning in Deep Neural Networks" mentions, it remains an open problem why neural networks find some "shortcut"
solutions easier to learn, compared to the "correct" solutions, but this is an interesting direction for future research.

**R2: Information bottleneck ad-hoc? Theoretical justifications?** The information bottleneck (IB) demonstrably
contributes to our method's performance (see paper Tab 2). Conceptually, our approach is built around identifying
observation histories as likely to contain nuisance information. The IB provides a natural way to penalize information
transmitted from this history. IB has been used in other works in similar ways, e.g. Pacelli 2020, "... Task-Driven
Control ...", and Rakelly 2018, "... probabilistic context variables". For theoretical justifications for IB, see Alemi et al
"Deep variational IB". We will motivate IB better in camera-ready and add these related works.

**R1: Error bars on action predictability.** The updated results with error bars are shown in the table below. We will
add error bars to Tab 3, 4, and 5 in camera-ready.

**R2: "Copycat" problem or just averaging out noise? Compare BC-SO?** Thank you for this perceptive comment.
BC-MO observes history, so it obtains full information and BC-SO is not comparable with it. We find it more reasonable
to compare with BC-SO (Full state) as it has the same information as BC-MO and, as suggested by R2, both of them
would suffer from the "averaging out". In the table below, we show that the next action is more predictable in BC-MO
than in the history-independent BC policy, suggesting that "copycat" problem exists in these environments. We will
clarify in camera-ready.

|  | Ant$\times 10^{-2}$ | Hopper$\times 10^{-3}$ | Humanoid$\times 10^{-1}$ | Reacher$\times 10^{-5}$ | Walker2d$\times 10^{-2}$ | HalfCheetah$\times 10^{-2}$ |
|---|---|---|---|---|---|---|
| expert | $6.91 \pm 0.21$ | $8.60 \pm 1.09$ | $6.93 \pm 0.32$ | $1.46 \pm 0.37$ | $2.47 \pm 0.07$ | $9.81 \pm 0.33$ |
| BC-SO (Full State) | $3.56 \pm 0.07$ | $2.55 \pm 0.72$ | $6.32 \pm 0.75$ | $0.33 \pm 0.05$ | $0.96 \pm 0.00$ | $3.32 \pm 0.11$ |
| BC-MO | $0.66 \pm 0.04$ | $1.07 \pm 0.16$ | $0.18 \pm 0.01$ | $0.32 \pm 0.05$ | $0.46 \pm 0.02$ | $2.97 \pm 0.15$ |

**R3: Figure 3 needs a line of best fit and $R^2$ value and explain the outliers.** We fit the curve with an inverse
proportional function, yielding $R^2 = 0.74$. To clarify, we do not claim that action predictability is the sole determinant
of reward, just one factor. Action predictability is a symptom of the copycat problem, but it is likely also influenced by
the nature of the specific task and demonstrations. As such, while the overall trends are clear, it is difficult to explain
outliers.

[Meta-Review · NeurIPS 2020]

Although the paper could do a better job of positioning itself with respect to prior work, there is a general consensus that the author’s rebuttal has addressed the major concerns and should be accepted.